# Effects of Strength Training Assessed by Anthropometry and Muscle Ultrasound

**DOI:** 10.3390/muscles4020011

**Published:** 2025-04-11

**Authors:** Juan Carlos Giraldo García, Gloria María Ruiz Rengifo, Donaldo Cardona Nieto, Julián Echeverri Chica, Juan Cancio Arcila Arango, German Campuzano Zuluaga, Oliver Ramos-Álvarez

**Affiliations:** 1Politécnico Colombiano Jaime Isaza Cadavid, Medellín 050021, Colombia; juangiraldo@elpoli.edu.co (J.C.G.G.); gloriaruiz@elpoli.edu.co (G.M.R.R.); donaldpf@gmail.com (D.C.N.); jcarcilaa@elpoli.edu.co (J.C.A.A.); 2GESTAS Research Group, Politécnico Colombiano Jaime Isaza Cadavid, Medellín 050021, Colombia; 3Sports Science Department, Universidad Pablo de Olavide, 41013 Sevilla, Spain; 4Health Sciences Department, University of Antioquia, Medellin 050010, Colombia; 5Health Sciences Department, University of CES, Medellin 050001, Colombia; 6Bioarray Genetic Diagnosis Laboratory, Bogotá 11013, Colombia; julianprotozoo@gmail.com; 7Clinical and Anatomical Pathology and Haematopathology, Clinical Haematological Laboratory, Medellín 050021, Colombia; gczuluaga@hematologico.com; 8Departamento de Educación, Área de Educación Física y Deportiva, Universidad de Cantabria, Los Castros Avenue, 50, 39005 Santander, Spain; 9Health Economics Research Group, Valdecilla Biomedical Research Institute (IDIVAL), 39011 Santander, Spain; 10Technology Applied to Occupational, Equality and Health Research Group (TALIONIS), Faculty of Health Sciences, University of A Coruña, de Oza As Xubias University Campus, 15006 A Coruña, Spain

**Keywords:** anthropometry, muscle ultrasound, strength training, muscle mass, echo-intensity, muscle thickness

## Abstract

Improving and maintaining an ideal body composition is important for sporting achievement and good health. Body composition assessment is therefore a tool used to monitor training and to evaluate the objectives of a training plan for health purposes. Ultrasound (US) emerges as an alternative to evaluate the thickness of subcutaneous cellular tissue, as well as muscle thickness: (1) Background: We aim to evaluate and compare the anthropometric and ultrasound measurements used to quantify the effects of strength training. (2) Methods: A total of 31 students (22.3 ± 4.14 years of age), 25 men and 6 women, from the Professional Programme in Sport were enrolled in the Physical Preparation course at the Institución Universitaria Politécnico Colombiano Jaime Isaza Cadavid. Protocol: Pre- and post-intervention ultrasound and anthropometric evaluations of a strength training programme with a predominance of the eccentric component were performed three times a week for 4 weeks. For the pre- and post-intervention relationship of the quantitative anthropometric and ultrasound variables, the Wilcoxon signed-rank test was used; the effect size of a Wilcoxon test was also calculated using the rank correlation, and the correlation of the anthropometric and ultrasound variables was determined using Spearman’s correlation coefficient, with a *p*-value < 0.05 considered statistically significant. (3) Results: There were no statistically significant differences in the anthropometric variables assessed, but there were significant differences in measures of quadricep muscle size and in the control parameter echo-intensity (EI) of subcutaneous fat in the variables. (4) Conclusions: The US of the quadriceps can measure changes in muscle thickness even without changes in muscle mass assessed by anthropometry, making it an excellent tool for the evaluation and monitoring of strength training.

## 1. Introduction

Improving and maintaining an ideal body composition is important for sporting achievement and good health. The assessment of body composition is therefore a tool used for training control and to evaluate the goals set in a training plan for health purposes. Different methods have been used to assess body composition. The first is the hydrostatic weight, which is considered the ‘gold standard’ but is limited by the time and cost of the equipment needed to perform it [1]. Dual X-ray absorptiometry is a highly accurate method that measures fat percentage, as well as bone mineral density [1]. The cost of the equipment and ionising radiation are limitations to its regular use. Bioelectrical impedance allows the measurement of body composition from a small electrical current sent through electrodes. It is easy and quick to apply but has the disadvantage of being highly variable in its results due to the multiple factors that can alter its measurement [1]. Anthropometry through fat folds, diameters and perimeters allows the evaluation of different components such as fat tissue, bone mass, lean mass and residual mass from equations that yield percentage values of the different components. It is the most widely used method due to its low cost, but its main limitations are that it requires good training, has a high inter-evaluator error and has difficulty in isolating adipose tissue from muscle tissue by palpation [1]. Muscle thickness is the simplest method to assess muscle size [2]. Ultrasound (US) has emerged as an alternative for assessing subcutaneous cellular tissue thickness, as well as muscle thickness. It is a low-cost method and, when compared to other imaging methods, does not emit ionising radiation. For these reasons, several studies consider US to be a valid method for assessing muscle size [3,4,5,6].

In addition, US provides information on muscle quality in terms of fat content by evaluating the intramuscular fat component on a grey scale, with the echo-intensity (EI) of subcutaneous fat used as a control parameter. Low EI values are associated with higher muscle quality [2], making EI a useful, economical and practical way to assess this factor [4]. Its validity has been corroborated by comparison with tomography (T) [7], muscle biopsy [8] and MRI. Echo-intensity not only assesses contractile proteins but also fat content, fibrous tissue and glycogen. Fat and fibrous tissue lead the image towards white (increased EI), while contractile proteins and glycogen lead it towards black (decreased EI). Therefore, a decrease in EI leads to a better-quality muscle, while an increase in EI indicates a deterioration of the muscle. Ageing and sedentary lifestyle are factors that can increase IE result from an increase in muscle fat content [5] and a decrease in contractile proteins. Therefore, EI can be a useful tool to assess the results of the application of training methods.

Based on all the previous elements, the aim of this study is to evaluate and compare anthropometric and ultrasound measurements used in the quantification of the effects of strength training.

## 2. Results

A total of 31 students of the Physical Preparation course, level VI of the Professional Programme in Sport of the Politécnico Colombiano Jaime Isaza Cadavid, were evaluated, corresponding to 6 women (19.35%) and 25 men (80.65%) who did not perform strength work in their daily training. The mean for age was 22.3 ± 4.14 years; for height, it was 1.73 mt; for weight, it was 70.46 kg; and for BMI, it was 23.13 kg/m^2^, corresponding to a normal classification according to the World Health Organization (WHO): 20 (64.5%) users were in this range, 9 (29%) were in pre-obese and 2 (6.5%) were in obese; the waist–hip index (WFI) recorded an average of 0.83 for the male gender and 0.76 for the female gender, with both classified as low cardiovascular risk according to the adaptation of the method of Berral (2011) [9] (Table 1).

Statistically significant changes were found in the categories defined by fat percentage (*p* < 0.05), decreasing the number of students in the medium category by 22.58% and increasing the good category by 19.4%, according to the formula differentiated for men and women by Jackson and Pollock [10] (Table 2), without significant changes in the fat percentage (*p* = 0.2184) (Table 3).

There were no statistically significant differences in the anthropometric variables evaluated. In the quadricep cross-sectional area ratio (CSSQ), according to the formula of Housh et al. (1995) [11], there is a variation in the estimation of muscle area as a function of thigh circumference and fat fold, and although it does not reach the reference value required to be significant, it could be related to an anthropometric improvement due to a decrease in the thigh skin fold caused by the strength work performed (Table 3).

There were statistically significant changes in measures of quadricep muscle size such as TRF (*p* < 0.01, t.e.: moderate), TAT (*p* < 0.01, t.e.: moderate), TVL (*p* < 0.05, t.e.: moderate), and TLT (*p* < 0.01, t.e.: large). On the other hand, there were significant changes in IE in the variables with internal control of fat IE, Dif1 (*p* < 0.0001, t.e.: large), Dif3 (*p* < 0.01, t.e.: large), and Dif4 (*p* < 0.0001, t.e.: large) (Table 4).

SFTTAR, as well as FTLT, had an excellent correlation with PG (R^2^ = 0.9, *p* < 0.01) and endomorphy (R^2^ = 0.774–0.787, *p* < 0.01) at both time points (pre- and post-intervention). TRF correlated with BM (R^2^ = 0.495–0.569, *p* < 0.01) and mesomorphy (R^2^ = 0.437–0.558, *p* < 0.05). TAT correlated with BM (R^2^ = 0.591–0.709, *p* < 0.01) and mesomorphy (R^2^ = 0.524–0.633, *p* < 0.01) (Figure 1).

## 3. Discussion

In the study conducted by Atencia et al. (2021) [12], the authors evaluated 56 university students of the Sports Science and Physical Activity Programme of the Corporación Universitaria del Caribe, finding values similar to the present study in terms of mean age (19.43 ± 2.23 years), weight (65.55 kg), height (1.70 mt), BMI (22.53), fat weight (FW) (11.65), muscle mass (31.55), fat percentage (17.37%), muscle percentage (48.24), and WHR (0.84). Differing in level, they belong to the first entry and they belong to the first year of the programme, with BMI of 22.53, fat weight of 11.65, muscle mass of 31.55, fat percentage of 17.37%, muscle percentage of 48.24, and a WHR of 0.84. Differing in level, they belong to the first entry and in the IPAQ, where 50% are in the moderate physical activity practice group and 23% and 26% in the low and high groups, respectively; this could be directly related to their progress in professional training, leading to greater practice of physical activity at a higher level [12].

Aravena et al. (2021) [13] interviewed university students with very similar characteristics to the present study; the initial sample was 30 Physical Education Pedagogy students (21 men and 9 women) from a private university in Chile, with no experience in muscle training, selected through non-probabilistic purposive sampling. The mean age was 19.97 ± 1.13 years (men: 20.00 ± 1.31 years and women: 20.00 ± 1.00 years), mean body weight was 66.95 ± 8.15 kg (men: 70.80 ± 7.46 kg and women: 61.28 ± 7.45 kg) and mean bipedal height was 169.50 ± 0.11 cm (men: 174.00 ± 0.07 cm and women: 161.00 ± 0.05 cm); the authors analysed 24 participants who completed the supervised muscle training programme for 8 weeks (16 sessions), 2 times per week, and were divided into two groups according to mean BMI: group 1 individuals were below it and group 2 individuals were above it. A significant decrease (*p* < 0.05) in adipose mass was evidenced in the subjects evaluated, and in the present study, an improvement in the anthropometric categorisation of fat percentage and no significant differences in body composition were found when comparing the groups, which differed in terms of the significant increase in muscle mass.

Cardozo et al. (2016) [14] carried out a characterisation of the body compositions of 82 university students from a sports faculty from the second semester of the academic sports performance programme in Bogotá, finding sociodemographic results similar to those of the current research; the mean age in men was 20.7 ± 2.3 years vs. 21.9 ± 1.3 years in women; the mean height was men 1.72 vs. women 1.59; the mean BMI was in a normal range, where 20% was in pre-obese and 3.5% lean and obese; and when contrasted with the present study, 64.5% (20 users) were found to be normal, 29% [9] pre-obese, and 6.5% [2] obese. In relation to the percentage of body fat in men (16.4) vs. women (25.0), both corresponded to a medium categorisation [14]. In contrast to the present study, they reported that in the stratification of fat percentage, there were statistically significant differences, reflected in a higher prevalence of overweight and obesity in women compared to the data found in men (46.67% vs. 20.9%).

The study presented by Rodríguez et al. (2020) [15] was carried out at Santo Tomás University, Tunja section, with 12 first-semester male students aged between 18 and 22 years; the intervention was performed for 10 weeks, 3 times a week, with 60 min per session: 5 min of warm-up, 25 min of strength exercises, 25 min of aerobic exercises, and ending with 5 min of muscle stretching; a classification of overweight grade II BMI (75%) and Type I obesity (25%) was found for BMI. In the post-intervention assessment, the findings of these two variables were similar to those of the present investigation: there were no significant changes in BMI despite decreasing the value, and there was a significant difference for fat % [15].

The cross-sectional area of the quadricep muscle was obtained by the equation validated by Housh et al. (1995) [11] based on anthropometric measures such as thigh fold and thigh girth, and although no significant differences were found, their variation was close to this benchmark. This indirect method could be considered a useful and easy way to apply a tool for evaluation in sport and clinical contexts at a low cost and with a longer intervention time, as happened in this study due to the abrupt end of the academic period, without having certainty over the level of sensitivity of the anthropometry at the muscular level. If an accurate estimation of the cross-sectional area of the quadriceps is required, imaging methods such as ultrasound and MRI are recommended.

Multiple studies that evaluated the effects of strength training, including plyometric training, used ultrasound as a method of assessing muscle mass by measuring muscle thickness. They demonstrated increased thickness of quadricep components as a result of training. The duration of these stimuli ranged from 8 to 16 weeks [16,17,18]. We found significant changes in only 4 weeks of training, which corresponds to those found in these studies. No changes were found as a result of eccentric dominated strength training on anthropometric variables.

A study conducted in obese patients who underwent bariatric surgery used rectus femoris thickness as a post-surgical control strategy, demonstrating benefits with respect to the use of DEXA and electrical bioimpedance, with which they showed an excellent correlation [19]. An excellent correlation was demonstrated between ultrasound-measured anterior thigh muscle thickness and quality of life [20], and it was therefore proposed as a measure that should be included in the diagnosis of sarcopenia [21].

Near-perfect correlations have been demonstrated between the assessment of anterior thigh muscle thickness by ultrasound compared to MRI (r = 0.99) [22]. Therefore, US is an excellent method to assess muscle size through muscle thickness [23]. The results of different studies showed a strong correlation between ultrasound and anthropometric measurements (r = 0.918) and DEXA (r = 0.848) to assess body composition [24,25]. It also provides higher measurement accuracy when compared to anthropometry [6]. In our study, we found significant differences in muscle thickness and subcutaneous cellular tissue assessed by ultrasound when there were no significant changes in anthropometric measurements of the same tissues, suggesting a greater sensitivity of ultrasound for assessing changes in both muscle thickness and subcutaneous cellular tissue as a measure of strength training control, especially when assessing the effects of exercise on muscle size or fat reduction.

In addition to muscle size, EI appears to be a strong predictor of functional outcomes from training or disuse in both healthy and diseased individuals [26]. EI can serve as an important clinical tool to assess the functional status of the musculoskeletal system, independent of age [27]. However, increased EI may be related to a decrease in muscle quality, muscle oedema as a sign of inflammation, or a decrease in muscle glycogen [28]. The study by Vasenina et al. (2022) in young adults assessed the effects on EI for a strength session immediately and at 24 and 48 h post-stimulus [29]. EI increased immediately but decreased significantly at 48 h. In our study, we observed a decrease in EI as a result of strength training with emphasis on the eccentric component in a period of only 4 weeks of intervention. These significant differences reflect changes that are quantified by EI, recognising that there are no concrete explanations for the physiological reasons for these changes. Longitudinal studies are needed to evaluate echo-intensity with muscle physio-logical variables to explain its modifications and therefore the role that it could play in the control of strength training.

## 4. Materials and Methods

### 4.1. Participants

A total of 31 students (22.3 ± 4.14 years of age), 25 males and 6 females, from the Professional Programme in Sport enrolled in the subject of Physical Preparation at the Institución Universitaria Politécnico Colombiano Jaime Isaza Cadavid participated in this study after signing the informed consent form and filling in the International Physical Activity Questionnaire (IPAQ). Each subject had to preserve their daily nutritional habits and not ingest liquor in the two days prior to the measurements. The following were the exclusion criteria: musculoskeletal injuries, cardiovascular disease, pharmacological and/or nutritional treatment, having started a training programme in the last 3 months, and not having signed and given informed consent. This study complied with the Declaration of Helsinki for research involving human subjects and was approved by the Ethics Committee of the Politécnico Colombiano Jaime Isaza Cadavid, with the file number #201801007381.

### 4.2. Procedure

The protocol consisted of a pre- and post-intervention ultrasound and anthropometric assessment of a strength training programme with a predominance of the eccentric component 3 times a week for 6 weeks. Due to a cessation of activities by the students, the intervention was carried out for only 4 weeks in order to guarantee their attendance at the post-intervention assessment. The pre-intervention ultrasound and anthropometric assessments were performed 4 days before starting the training programme and the post-intervention assessment was performed 4 days after the last stimulus to avoid the acute effects of exercise. During the intervention the students performed their usual academic activities, excluding the strength stimuli.

### 4.3. Strength Training

The training programme was designed to last 6 weeks, planning the load based on references such as Schoenfeld (2010), who indicates that with constant training for 6–10 weeks there will be hypertrophy with eccentric training [30], and in some cases changes can be noticed after 4 weeks, but the most significant results are usually seen after 8–12 weeks [31]. The number of weekly sessions recommended for muscle fibre hypertrophy by Schoenfeld (2016) [32] are 2–3, with 4–6 sets and 4–8 repetitions with 80–100% of the eccentric 1RM (with a higher load than would be used in the concentric phase) [33], compared with a time under tension of 3–6 s in the eccentric phase and the recovery time after an intense eccentric stimulus of 48–72 h according to Franchi et al. (2017) [34], and Proske and Morgan (2001) mention that the effects of muscle damage induced by eccentric exercise can persist for up to 48 h, recommending an interval of at least that long to favour optimal muscle regeneration [35].

In this study, we were able to intervene with eccentric training for 4 weeks with 12 sessions, instead of 6 weeks with 18 sessions, as was the initial protocol, given the novelty of the cessation of activities by the university students. A mesocycle was developed, in which each microcycle contained three weekly sessions of eccentric training of the lower body and the central zone and one of active recovery with regular sport for a total of 12 sessions, held on Monday, Wednesday and Friday at 10 am for approximately 45 min. Each training session included a 10 min warm-up with fundamental low-intensity exercises involving the core zone, with the aim of warming up and providing control of the core zone. The load structure in terms of % of RM, sets, and repetitions is described in Table 1, validating the protocol with the respective author. Each phase contained the number of days, % of work, sets, and repetitions, starting with phase 1 for 6 sessions of Neuromuscular Adaptation and improvement of technique [36]; phase 2 of Hypertrophy and Progressive Load, with increased volume and mechanical tension [30]; phase 3 of Muscle Hypertrophy; and phase 4 of Maximum Intensity and Strength (not applied), with the objective in phases 3 and 4 being to train high-threshold motor units [37]. The total time under tension (TUT) was determined according to the methods of Benavides-Villanueva and Ramírez-Campillo (2022) [38] and Wilk et al. (2021) [39]; bearing in mind that the optimal time per repetition to obtain hypertrophic adaptations is in the range of 2 to 6 s, we opted for a slow eccentric phase of 4 to 6 s and a fast concentric phase of 1 to 2 s, with a total TUT per set of between 30 and 70 s, according to the number of repetitions. Some basic exercises were performed such as eccentric squats (descent in 4–6 s), eccentric Romanian deadlifts (slow and controlled descent), bench presses with eccentric emphasis (descent in 5 s), and eccentric pull-ups (descent in 6 s), planks, and box jumps (descent in 6 s). The rest time between sets was determined according to the method of Mazzetti et al. (2000) [40]: for phase 1 and 3, the time was 60–120 s; in phase 2, it was 45–90 s; and for phase 4, it was 60–150 s. Table 1 details the characteristics of the work carried out on different days (Table 5).

### 4.4. Anthropometric Measurements

ISAK anthropometry is a standardised methodology for measuring body composition, including muscle, fat, and bone mass. It requires plicometers, tape measures, and bone calipers. The ISAK is an indispensable tool for planning sports training programmes. By identifying and analysing these measurements, the trainer can adjust routines to achieve goals, improve performance, and prevent injury.

Basic measurements of body composition were taken, such as weight by means of a Tanita Hd-314 electronic scale, expressed in kilograms, and height with a portable wall height measuring device with a range of 0 to 200 cm and a precision of 1 mm, recorded in metres; with these two data measurement tools, the body mass index (BMI) was obtained per person and interpreted according to the WHO classification. For the assessment of skin folds, a calibrated Slim Guide adipometer was used; perimeters were measured with a tape measure and diameters with Tacklife digital Vernier or King’s foot tools. Obtaining the different components such as fat tissue, bone mass, lean mass, residual mass, and somatotype under the methodology of the ISAK and to identify cardiovascular risk, the waist–hip circumference was evaluated and interpreted by the parameters given by the WHO (2008) [41].

### 4.5. Description of the Taking of the Main Anthropometric Measurements

Weight: The mass of the body expressed in kilograms, taken on a scale. Subjects were assessed in a standard upright position, with their backs to the measurement, in the centre of the scale while wearing a minimal amount of clothing, such as shorts and a T-shirt.

Size: The distance between the vertex and the heels, recorded in centimetres by means of a tape measure in the vertical plane of the measuring rod. The measurement was taken from each subject in a vertical position, looking straight ahead and after a deep breath.

BMI: The ratio of weight in kilograms divided by the square of height in metres, by means of which the subject was categorised as underweight, a healthy weight, overweight, or obese. It is an indirect measure that is used as a screening measure to indicate body fat and health risk.

WHI (Waist–hip index): The correlation between the waist circumference located in the area of the smallest circumference of the abdomen, midway between the costal border and the iliac crest, and the hip located in the widest area of the pelvis, approximately at the level of the greater trochanter. This measurement is associated with risk of cardiovascular disease and disorder [42,43].

Skinfolds: These refer to the amount of subcutaneous adipose tissue without including the muscle. They are measured using a calliper and expressed in millimetres (mm). The eight skinfolds of the right side of the body were assessed according to the ISAK protocol, namely subscapular, triceps, biceps, iliac crest, suprailiac, abdominal, thigh, and medial calf.

Perimeters: These are the circumferences of specific areas, expressed in centimetres (cm) and measured using a measuring tape. According to the ISAK manual, the recorded measurements included the relaxed and contracted arm, minimum and maximum waist, and maximum thigh circumference.

Diameters: This refers to the distance between two established anatomical points at the level of bony prominences. The instrument used was a caliper, and measurements were recorded in centimetres (cm). The ISAK protocol specifies measuring the biepicondylar diameter of the humerus and femur and the bistyloid diameter of the wrist.

### 4.6. Ultrasound Measurements

Transverse and longitudinal images were obtained from the quadriceps femoris on the right limb using a B-mode ultrasound system (B-Ultrasonic Diagnostic System, Contec, CMS600P2, Hebei, China). A linear transducer (gain: 58; frequency: 7.5 MHz; depth: 6 cm), covered with a sufficient amount of water-soluble transmission gel to prevent compression of the skin surface, was placed perpendicular to the longitudinal and transverse axes of the quadriceps femoris. The transducer was positioned at the midpoint between the anterior superior iliac spine and the superior pole and between this point and the superolateral angle of the patella for anterior and lateral images, respectively. Subjects were evaluated in the supine position, having rested for at least five minutes and without engaging in vigorous physical exercise on the same day. Two longitudinal-section images and two transverse-section images were taken at each midpoint. The frozen image was digitised and subsequently analysed using the open-source software ImageJ (National Institute of Health, Bethesda, MD, USA, version IJ 1.46). The anterior transverse-section images were used to measure rectus femoris muscle thickness (from inferior margin of the anterior fascia of the rectus femoris to the superior margin of the posterior fascia of the rectus femoris), vastus intermedius thickness (inferior margin of the intermuscular fascia to the femoral periosteum) and total anterior quadriceps thickness (from the inferior margin of the rectus femoris to the femoral periosteum). The lateral transverse-section images were used to measure vastus lateralis muscle thickness (from the inferior margin of the anterior fascia of the vastus lateralis to the superior margin of the posterior fascia of the vastus lateralis), vastus intermedius thickness (lateral view) (from the inferior margin of the intermuscular fascia to the femoral periosteum), and total lateral quadriceps thickness (from the inferior margin of the vastus lateralis to the femoral periosteum). The transverse-section images were also used to determine the echo-intensity (EI) of the evaluated muscles using the histogram function in ImageJ. The region of interest was selected as the largest rectangular area of each muscle, excluding fascia. The mean value of the two images was expressed as a value between 0 (black) and 255 (white). EI correction was performed based on the thickness of the subcutaneous tissue, as proposed by Young, and the fat percentage was measured using the method proposed by the same author for all muscles [44]. Additionally, as a control strategy, the difference in the EI of the fat relative to each evaluated portion of the quadriceps was calculated, corresponding to Dif1 to Dif6 [45]. The longitudinal-section images were used to determine the pennation angle of the rectus femoris and vastus lateralis. The values used for statistical analysis for muscle thickness and pennation angle were the averages of the two measurements from each image.

### 4.7. Statistical Analysis

For the descriptive analysis of anthropometric and ultrasound aspects, absolute distributions, relative distributions, and summary indicators such as the median and the median absolute deviation were used. The normality criterion was established using the Shapiro–Wilk test.

For the comparison of quantitative anthropometric and ultrasound variables before and after the intervention, the Wilcoxon signed-rank test was used. Additionally, the effect size of the Wilcoxon test was calculated using rank correlation. A small effect was considered for values between 0.1 and 0.3, a moderate effect for values between 0.3 and 0.5, and a large effect for values greater than 0.5.

The correlation between anthropometric and ultrasound variables was determined using Spearman’s correlation coefficient. A *p*-value < 0.05 was considered statistically significant.

## 5. Conclusions

Our results suggest that quadricep ultrasound can measure changes in muscle thickness even in the absence of changes in muscle mass assessed through anthropometry, making it an excellent tool for evaluating and monitoring strength training. Additionally, EI may serve as an important clinical tool for assessing the functional state of the musculoskeletal system and, in particular, muscle adaptations to strength training. Therefore, ultrasound becomes an easily accessible, non-invasive, cost-effective, and more sensitive method compared to anthropometry, which could provide valuable information on changes in body composition (fat and muscle components) for training monitoring.

### 5.1. Limitations

The low participation of women did not allow a comparison of results between sexes, but we did not consider it appropriate to exclude them from the analysis, although their physiological differences may influence the results. Suggesting no alcohol consumption for only 48 h before testing is a limitation. It was not possible for us to strictly control consumption, which may influence the results, with consumption close to the tests but longer than 48 h beforehand having its potential effects on strength. The abrupt cessation of the academic period could affect the results of the anthropometric variables, preventing us from being able to determine whether indirect anthropometric measurements can detect changes in body composition with a more prolonged intervention, considering that ultrasound is sensitive and reliable for muscle assessment.

### 5.2. Suggestions

Longitudinal studies are required to evaluate echo-intensity with muscle physiological variables to explain its modifications and therefore the role that it could play in the control of strength training. In addition, it is recommended to use the indirect method of quadricep cross-section referred to by Housh et al. (1995) [11] in longer interventions and to contrast these measurements with ultrasound to define its sensitivity, as it could be a good alternative for the muscle control of athlete and is easily accessible, reliable and low cost.

## Figures and Tables

**Figure 1 muscles-04-00011-f001:**
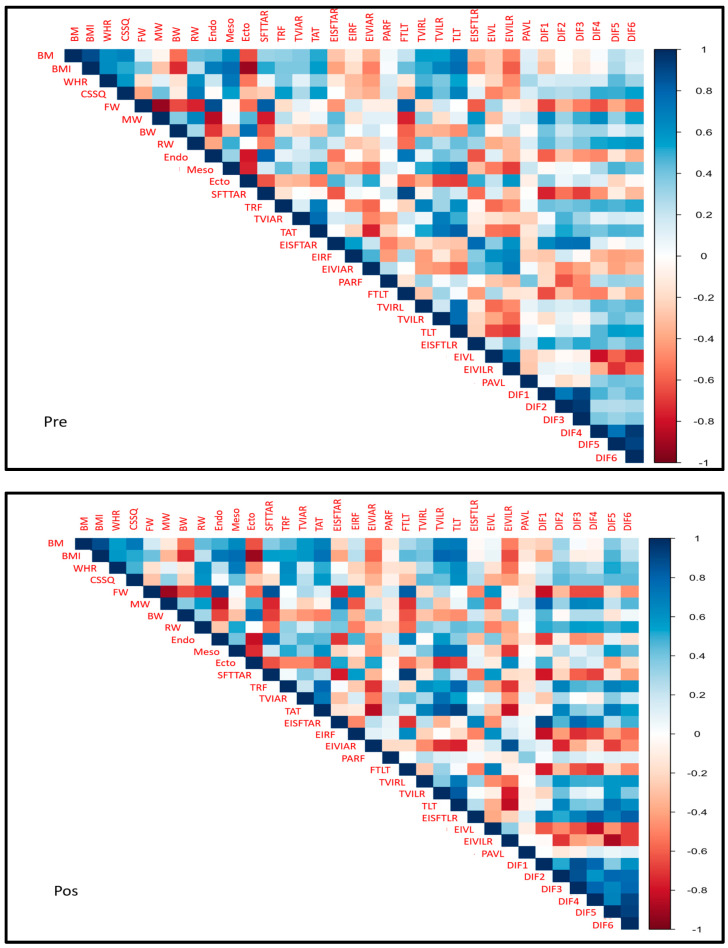
Correlations between anthropometric indicators and ultrasound findings.

**Table 1 muscles-04-00011-t001:** Descriptive statistics.

Descriptive Statistics	Data
n ^1^	31
Age (years) (M ± SD) ^1^	22.3 ± 4.14
Height (meters) (M ± SD) ^1^	1.73 ± 0.08
Body Mass (Kg)(Mediana ± DAM) ^1^	69 ± 13.64
BMI (Kg/m^2^) (Mediana ± DAM) ^1^	23.54 ± 3.34

^1^ Note. M: mean. SD: standard deviation. DAM: absolute deviation from the median. n: sample size. BMI: body mass index.

**Table 2 muscles-04-00011-t002:** Distribution of fat percentage interpretation.

Anthropometry (% Fat)	Pre- n (%)	Post- n (%)	*p*-Value
Media	21 (67.74)	14 (45.16)	0.0302 ^1^
Good	1 (3.2)	7 (22.6)	
Excellent	9 (29.03)	10 (32.25)	
Total	31 (100)	31 (100)	

^1^ Chi-square test of marginal homogeneity: the Stuart and Maxwell extension to McNemar’s test. Significance at *p* < 0.05.

**Table 3 muscles-04-00011-t003:** Distribution of anthropometric features.

Anthropometry	Before		After		*p* Value *	Size of the Effect
	Medium	DAM	Medium	DAM		
BM	69.00	13.64	70.00	13.34	0.27	0.19
BMI	23.54	3.34	23.13	3.76	0.27	0.19
WHR	0.82	0.06	0.83	0.04	0.78	0.051
CSSQ	74.88	7.61	72.24	10.16	0.05	0.34
FW	14.04	5.84	14.25	5.92	0.21	0.22
MW	46.49	4.05	46.27	4.12	0.39	0.15
BW	15.53	2.42	15.66	2.27	0.26	0.20
RW	24.10	0.00	24.10	0.00	n.a.	n.a.
Endo	5.30	1.33	5.50	1.78	0.13	0.27
Meso	4.70	1.63	4.80	1.48	0.47	0.13
Ecto	1.80	1.48	1.90	1.48	0.29	0.19

Note: Test ranges of the Wilcoxon test. * Significance *p* < 0.05. DAM: absolute deviation from the median. BM: Body mass. BMI: Body mass index. WHR: Waist-to-hip ratio. CSSQ: Cross-sectional section of the quadriceps. FW: Fat weight. MW: Muscle weight. BW: Bone weight. RW: Residual weight. Endo: Endomorphy. Meso: Mesomorphy. Ecto: Ectomorphy. n.a.: Not applicable.

**Table 4 muscles-04-00011-t004:** Distribution of the aspects of Ultrasound.

Ultrasound	Medium	DAM	Medium	DAM	*p* Value *	Size of the Effect
	Pre		Post			
SFTTAR	5.120	3.050	5.690	3.560	0.827	0.041
TRF	25.930	4.050	27.370	5.460	0.004 **	0.500
TVIAR	20.450	2.940	21.000	3.460	0.158	0.255
TAT	48.060	6.910	49.080	7.410	0.005 **	0.484
EISFTAR	155.220	17.010	157.160	11.970	0.111	0.289
EIRF	116.290	18.160	112.050	9.610	0.182	0.243
EIVIA	90.050	15.880	90.380	13.670	0.999	0.000
PARF	15.120	4.560	17.000	2.620	0.055	0.345
FTLT	5.060	3.230	5.600	3.290	0.812	0.044
TVL	24.320	4.180	25.120	4.040	0.035 *	0.377
TVILR	20.400	4.630	19.510	4.370	0.820	0.042
TLT	47.590	7.990	48.580	6.520	0.003 **	0.516
EISFTLR	151.590	10.900	155.480	9.980	0.135	0.271
EIVL	119.740	18.020	117.390	10.780	0.157	0.257
EIVILR	78.470	14.620	80.030	19.120	0.399	0.155
PAVL	16.130	3.590	15.650	3.190	0.961	0.011
Dif1	33.000	15.540	41.170	18.320	<0.0001 **	0.686
Dif2	67.170	15.400	71.730	21.280	0.224	0.222
Dif3	49.110	14.060	55.630	13.730	0.002 **	0.524
Dif4	30.780	24.940	37.920	19.890	0.001 **	0.560
Dif5	79.520	13.800	84.100	23.020	0.854	0.035
Dif6	56.650	19.270	59.770	21.540	0.157	0.257

Note: The Wilcoxon signed-rank test. * Significance *p* < 0.05; ** significance *p* < 0.01. DAM: absolute deviation from the median. SFTTAR: anterior region subcutaneous fatty tissue thickness. TRF: RF thickness. TVIAR: LV thickness in the anterior region. TAT: total anterior thickness. EISFTAR: EI anterior subcutaneous fatty tissue. EIRF: RF EF: RF EF. EI-VIA: EI of the VL in the anterior region. PARF: RF pennation angle. FTLT: lateral region fat thickness. TVL: VL thickness. TVILR: VL thickness in lateral region. TLT: lateral total thickness. EISFTLR: EI of subcutaneous fatty tissue in the lateral region. EIVL: VL IE: VL IE. EIVILR: EI of the VL in the lateral region. PAVL: VL pennation angle. Diff1: anterior fatty IE minus RF IE. Diff2: EI anterior fat minus EI of the anterior LV. Diff3: EI anterior fat minus average EI of RF and EI of anterior VL. Diff4: EI Lateral fat minus EI of VL. Diff5: anterior IE fat minus lateral VL IE. Diff6: lateral IE Lateral fat minus average VL IE + lateral VI IE.

**Table 5 muscles-04-00011-t005:** Load progression.

Stage	Days	% 1RM	Series	Repetitions	Reference
1	1–6	60–70	4–5	10–12	Bompa and Carrera (2015) [16]
2	7–9	70–80%	4–5	8–10	Schoenfeld (2010) [8]
3	10–12	80–90%	5–6	5–6	Zatsiorsky and Kraemer (2020) [15]
4	13–18 *	85–95%	4–5	4–5	Verkhoshansky (2009) [17]

Note: 1RM: one maximum repetition. * Not carried out due to the cessation of academic activities.

## Data Availability

The data presented in this study are available on request from the corresponding author.

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
