# Peer review of "Effects of Strength Training Assessed by Anthropometry and Muscle Ultrasound"

_muscles, 2025, doi:10.3390/muscles4020011_

Round 1
Reviewer 1 Report
Comments and Suggestions for Authors
Dear Authors,
Thank you for the opportunity to review your manuscript. The study addresses a relevant topic and presents interesting data. However, I have identified some aspects that could be improved to increase methodological clarity and interpretation of the results.
Below is a detailed review with specific suggestions.
Abstract
Lines 27-28: It is unclear whether participants had previous experience with strength training. There is no specification whether they were already engaged in regular training programs or were sedentary. Additionally, there is no mention of specific selection criteria (e.g., no previous medical conditions, no use of supplements, etc.).
Lines 39-40: There is a lack of reference on the validity of ultrasound compared to other methods for assessing muscle mass.
Lines 45-49: The description of body composition assessment methods (hydrostatic weighing, DXA, BIA) is helpful, but a clearer transition to ultrasound as the preferred method would be helpful.
Lines 54-60: The criticism of bioelectrical impedance analysis (BIA) is valid, but it should be noted that its reliability depends on standardized measurement conditions.
Lines 67-71: A recent bibliographic reference or supporting data should be added.
Lines 74-76: It would be useful to specify whether this study has an applied relevance (e.g., athlete monitoring).
Results
Lines 81-83: The BMI classification of the subjects follows WHO criteria, but a comparison with similar studies would help contextualize the sample.
Lines 90-92: The authors report a significant reduction in body fat percentage among participants. However, they do not specify the exact percentage of reduction (how much did it decrease from the initial value?). The statistical test used to determine significance is not explicitly mentioned, and while the p < 0.05 value is reported in the table, it should also be included in the text. Moreover, it is unclear whether the reduction was uniform across men and women or if differences existed between groups.
Lines 97-99: The authors state that the cross-sectional area of the quadriceps (CSA) varied following training, but the current wording is ambiguous because it is unclear:
Whether the variation is statistically significant (p < 0.05 or p > 0.05?).
Whether the variation is clinically relevant, meaning if the change has a real impact on muscle function or is merely a numerical fluctuation.
Whether the change is uniform across the sample or if differences exist between men and women.
Specifically, it is not clearly explained:
The extent of the variation (in cm² or % compared to the baseline?).
How close the value is to the threshold of significance (if p = 0.06, it suggests a trend, while if p = 0.3, it is likely irrelevant).
Whether this variation has practical implications for athletes or muscle function.
Lines 100-104: The effects on ultrasound variables are well described, but the level of significance varies among measures. It is recommended to discuss the practical relevance of these variations in terms of performance, injury prevention, or rehabilitation.
Discussion
Lines 143-146: The authors cite Cardozo et al. (2016) to discuss obesity differences between men and women. However, this information is not directly related to the study's results. The present study does not specifically analyze obesity differences by gender. Additionally, it is unclear why this citation is emphasized since the study focuses on ultrasound and the effects of eccentric strength training. The key question is: Why emphasize the relationship between obesity and gender? There is no explanation of how this difference impacts the studied variables. For example, did gender differences influence the ultrasound measurements?
Lines 160-161: The manuscript states: "No changes were found in anthropometric variables following eccentric strength training." However, there is no discussion on why ultrasound detected differences while anthropometric measurements did not.
Lines 169-172: The comparison between US and MRI is well documented, but a more recent reference specifically addressing the precision of ultrasound compared to MRI in a sports context would strengthen this section.
Methods
The Materials and Methods section should be presented before the Results, as it provides the methodological context necessary to understand how the data were obtained.
Lines 193-198: The exclusion criteria are well defined, but the restriction on alcohol consumption two days before measurement may be insufficient to eliminate metabolic effects. Studies indicate that alcohol’s effects on muscle parameters can last several days (up to 5-7 days, depending on the amount consumed). If testing was conducted only 48 hours after the last alcohol intake, there is a risk that alcohol may still influence ultrasound results, potentially biasing conclusions. This could be a limitation of the study.
Sample imbalance (25 men, 6 women): Could this affect the results?
Lines 205-208: The training protocol was initially planned for 6 weeks but was reduced to 4 weeks due to the cessation of university activities. This modification may have influenced the results and should be emphasized more strongly and explicitly acknowledged as a study limitation.
Lines 215-218: The reference to Schoenfeld (2010) is valid, but details on load progression within the actual 4-week training period are missing.
Lines 231-236: The article describes the eccentric training protocol (e.g., number of sets, repetitions, and load used), but it does not specify the total time under tension (TUT) for each exercise. TUT is a fundamental parameter in strength training, particularly for muscular adaptations.
It is well established that eccentric training is time-dependent, as the duration of the eccentric phase influences muscular and neural adaptations.
A longer TUT (e.g., 3-5 seconds per repetition) increases mechanical tension and metabolic stress, stimulating hypertrophy and structural adaptation.
If TUT is too short, the effects of eccentric training may be minimal or suboptimal.
This is crucial for comparing the results with other studies.
If TUT is not indicated, it is difficult to replicate the study or compare it with previous research.
For example, a protocol with 4-second eccentric phases will have different effects than one with a 1-second phase.
The impact on ultrasound and anthropometric measures should also be considered. If the TUT was short, this could explain the lack of significant changes in anthropometric variables compared to ultrasound results. Ultrasound may have detected more subtle structural adaptations that a longer TUT protocol would have amplified.
Conclusions
Lines 341-344: The statement “EI can serve as an important clinical tool for assessing the functional status of the musculoskeletal system” is strong, but should be supported by better contextualization with the study results.
Lines 347-348: The conclusion that ultrasound is a more sensitive method than anthropometry is clear, but consideration of the limitations of the study is lacking
Author Response
Dear Reviewer:
I hope you are well. We have proceeded to implement all the suggestions you have provided and uploaded a new version of the document. I hope it fits. Best regards.
The authors

Reviewer 2 Report
Comments and Suggestions for Authors
The text is well-structured from a scientific perspective, and I believe it is an interesting and appropriate manuscript for this journal.
However, I think it would benefit from a revision to improve its fluidity and readability, making it more natural and engaging for readers.
The results section is very technical and rich in data, which is positive, but I think it could be easier to read by adding a summary sentence after each data block to emphasize the main findings. I believe this would also help readers who are less experienced in this type of scientific research and might be studying this manuscript. Additionally, it would make this section more organized.
Regarding the explanation of the results and the subsequent discussion, I think it is very well done. However, in these sections, I have some questions for the authors that could improve the completeness of the manuscript.
I would like to ask whether it is possible to clarify better whether the lack of anthropometric variations alongside ultrasound variations indicates a greater sensitivity of ultrasound in detecting early muscle changes.
Furthermore, from the text, I understand that ultrasound has an "excellent correlation" with DEXA, bioelectrical impedance, and MRI, but no numerical correlation value (r) is provided. I would suggest specifying briefly which ultrasound parameters show the strongest correlation with the other techniques in this part of the discussion.
I would also like to ask if it is possible to explain more clearly how a reduction in Echo-Intensity (EI) indicates an improvement in muscle quality, associated with a decrease in intramuscular fat content and an increase in myofibrillar density.
In conclusion, I think it would be interesting to briefly discuss whether the change in EI is clinically relevant or if further research is needed to establish a significant threshold.
The last small issue I would like to point out to the authors is to check how "Echo-intensity" is written. In some parts, it is written as "Echo-intensity," while in others, it appears as "Eco-intensity."
After these minor improvements, I believe the editors could consider its publication in Muscles.
Comments on the Quality of English LanguageThe text is well written; however, there are small aspects that could be improved, which I have suggested in the revisions.
Author Response

(The authors gave the same response as above.)

Round 2
Reviewer 1 Report
Comments and Suggestions for Authors
Dear Authors,
I have reviewed the revised version of your manuscript and confirm that all previous comments have been addressed appropriately and thoroughly.
The work is now clear, complete, and methodologically sound.
I have no further remarks.
Best regards,
Reviewer 2 Report
Comments and Suggestions for Authors
I thank the authors for positively accepting my suggestions. I believe that the revisions they made were thorough and effectively explained the responses to my requests. For this reason, I give a very positive final opinion.